# Design and Implementation of a Video-Frame Localization System for a Drifting Camera-Based Sewer Inspection System

**DOI:** 10.3390/s23020793

**Published:** 2023-01-10

**Authors:** Yusuke Chikamoto, Yuki Tsutsumi, Hiroaki Sawano, Susumu Ishihara

**Affiliations:** 1Graduate School of Integrated Science and Technology, Shizuoka University, Hamamatsu 432-8011, Japan; 2Department of Information Science, Aichi Institute of Technology, Toyota 470-0392, Japan; 3College of Engineering, Academic Institute, Shizuoka University, Hamamatsu 432-8011, Japan

**Keywords:** sewer inspection, sensor network, localization, drifting wireless camera, wireless LAN, image processing

## Abstract

To reduce the cost of inspecting old sewer pipes, we have been developing a low-cost sewer inspection system that uses drifting wireless cameras to record videos of the interior of a sewer pipe while drifting. The video’s data are transmitted to access points placed in utility holes and further transmitted to a video server where each video frame is linked to its capturing position so that users can identify the damaged areas. However, in small-diameter sewer pipes, locating drifting nodes over the full extent of the pipeline using Wi-Fi-based localization is difficult due to the limited reach of radio waves. In addition, there is the unavailability of a GNSS signal. We propose a function to link each video frame to a position based on linear interpolation using landmarks detected by the camera and image processing. Experiments for testing the accuracy of the localization in an underground sewer pipe showed that all utility holes were successfully detected as landmarks, and the maximum location estimation accuracy was less than 11.5% of the maximum interval of landmarks.

## 1. Introduction

### 1.1. Background: Sewer-Pipe Inspection with Drifting Wireless Cameras

Aged sewer pipes are susceptible to severe problems such as clogging by sand entering from cracks or by tree roots; the deterioration of sewer function; and road collapse. The year 2022 marks more than 30 years since the installation of 40% of a total of 490,000 km of sewer pipes in Japan [1,2]. Therefore, it is necessary to inspect aging sewer pipes regularly and repair or replace any damaged parts. However, current sewer-pipe-inspection methods, such as visual inspection [3], fiberscope inspection [4], self-propelled robot inspection [5], and boat-type camera inspection [6], are costly and/or time-consuming.

To shorten the time and reduce the monetary costs, we propose a sewer-pipe-inspection method using drifting wireless camera nodes (CNs) [7], as shown in Figure 1. When inspecting a sewer pipe with this system, an inspector firstly places one or more CN into the sewer pipe. Then, the CNs start recording videos of the interior of the pipe. When each of the CNs reaches the wireless communication range of one of the access points (APs) installed at some utility hole, it sends the recorded video data to the AP via wireless LAN. Finally, a server in the cloud aggregates the data from the APs via cellular networks. Then, users, i.e., sewer inspectors, access the server and browse the video to identify the positions of damage.

This system is safe because inspectors do not need to enter the pipes. In addition, unlike fiberscopes and self-propelled robot methods, the inspection range is not limited by the cables. Furthermore, unlike the boat-shaped camera technique, it is possible to watch the captured images before the camera drifts downstream through the sewer pipe. It is also possible to identify where the video frame was captured. These features reduce labor costs and are convenient.

This system uses wireless video data transmission from CNs in a sewer pipe. To identify the performance of the wireless communication in narrow (ϕ200–250 mm) sewer pipes, which are the most widely used public sewer pipes in Japan, we conducted an experiment. Moreover, we designed a wireless communication protocol for a narrow underground pipe environment. In [8], we evaluated the performance of the wireless communication according to IEEE 802.11 and IEEE 802.15.4 at 2.4/5 GHz and ARIB STD-T108 at 920 MHz in a 200–250 mm in diameter underground sewer pipe. The measurement results revealed that the maximum communication distance is approximately 10 m in the sewer pipe, whereas the interval between utility holes is typically 30 m or more. Ishihara et al. [7] developed a protocol to collect video data from multiple CNs to compensate for the short wireless communication range.

### 1.2. Motivation: Difficulty of Localization in Sewer Pipes

Even when video data can be reliably collected via wireless communication in the neighborhood of APs, identifying the positions of pipe damage, such as cracks and clogged tree roots from the captured video, is difficult because it is unclear where the video was taken. To detect the positions of pipe damages by looking at the aggregated video data, it is necessary to identify where each video frame was taken. However, GNSS cannot be used in sewer pipes, and location estimations of CNs using radio waves are also difficult because the range of the radio wave’s communication in a narrow pipe (diameter of 200–250 mm) is limited. Additionally, the CN’s weight should be light so that it can easily drift down a sewer pipe. Thus, the additional devices for the location estimation should be minimum. To tackle this problem, it is necessary to link each video frame to the position where it was captured with the minimum additional devices and without depending on radio communication.

### 1.3. Contributions

The contributions of this paper are summarized as follows. Firstly, we propose a method to estimate CN positions based on linear interpolation using utility holes and pipe joints as landmarks in a sewer pipe. We also implemented the proposed method together with a browsing system that links location information to each frame of the video and conducted experiments to evaluate the accuracy of the proposed localization function using a drifting CN in a real underground pipe.

### 1.4. Structure of the Paper

The remainder of this paper is structured as follows: we introduce related work on the self-position estimation of wireless mobile devices in Section 2. In Section 3, we describe a sewer-pipe-inspection method using drifting wireless cameras that we developed. In Section 4, we propose a method to link the location of a CN to a timestamp of a video frame. Section 5 describes the implementation of the proposed localization system, and Section 6 describes experiments for the verification of the system in an underground pipe. Section 7 summarizes this paper.

## 2. Related Work

In this section, we review studies on indoor localization methods, particularly focusing on their applications in sewer pipes. To link each of captured video frames to the position of a CN, it is necessary to estimate the CN’s position at a given timestamp. Generally, GNSS and signal strength from wireless LAN and cellular phone base stations are used to estimate the locations of wireless communication devices outdoors. Radiowaves from satellites and base stations, however, do not reach the interior of sewer pipes. Therefore, it is necessary to develop self-localization techniques for CNs that do not rely on them. By considering the utilization of a sewer inspection system with drifting cameras moving in a narrow pipe, the weight of the equipment should be small, and the number of additional devices needed to be installed in the environment should be minimal. Potential methods for localization of CNs in sewer pipes include using visible light, sound waves, radiowaves, inertial measurement units (IMUs), and cameras.

### 2.1. Visible Light

Visible light enables communication over 1 km using 400–800 THz light waves, which are modulated and emitted mainly by light-emitting diodes (LEDs) [9]. Visible-light-based localization is generally used with an image sensor attached to a mobile device to receive signals from recognized multiple light-emitting sources installed in an indoor facility. The receiver estimates its self-position together with the location of the sources. Zhang et al. [10], Li et al. [11], and Guan et al. [12] proposed methods for localizing one light receiver with 1 or 2 light-emitting sources installed in a ceiling and achieved a centimeter-level positioning accuracy. The main drawbacks of their localization technologies are that the available distance between the emitter and receiver is limited by the strength of the light and the directions of the emitter and receiver. Since CNs rotate while drifting, the control of light directions is difficult. In addition, if an obstacle in sewer pipes blocks light from sources, the receiver cannot recognize the light, resulting in a significant localization error. Furthermore, the weights of the source and receiver are so heavy that it is difficult to mount them on a drifting CN.

### 2.2. Sound Waves

Sound-wave-based localization can be used to estimate the position of CNs in sewer pipes. Sound waves are relatively slower than radio waves; thus, it is easy to measure the arrival delay of sound waves, enabling localizations with high accuracy (error of a few centimeters). There are two types of sound wave localization: ultrasonic and acoustic.

The range of ultrasonic-wave-based localization is generally about ten meters in the air [13]. Chew et al. [14] proposed ultrasonic-ToF-based localization methods using a moving receiver and multiple ultrasonic transmitters mounted on a ceiling of a room to estimate the position of the receiver moving on a desk. They achieved several-centimeter-scale accuracies of the localization of the receiver. Hoeflinger et al. proposed ultrasonic-ToF-based localization methods using an ultrasonic transmitter and multiple receivers placed on a device [15] in an indoor environment, and the device estimated a target position with an error of a few centimeters. Zhang et al. proposed TDOA-based localization methods using a moving ultrasonic transmitter and multiple time-synchronized receivers placed at several areas in the room [16]. They located the moving transmitter with an error of a few tens of centimeters. Our proposed inspection system assumes that the CNs move over long sections of several hundred meters or more; thus, ultrasonic-based-localization is unsuitable in our proposed sewer inspection system.

Since acoustic waves generally have less wave attenuation than ultrasonic sound, they propagate over longer distances than ultrasonic sound. Worley et al. proposed ToF-based localization utilizing acoustic echo in a 15 m pipe using a loudspeaker and microphone installed on a robot [17]. They achieved localization with an error level of a few tens of centimeters. However, acoustic sound waves are susceptible to noise, such as the sound of flowing or falling water in a sewer pipe. In addition, since pipes are connected to various buildings, the loudness of other sounds can lead to noisy data. Therefore, acoustic-sound-based localization is unsuitable in our sewer proposed inspection system as well.

### 2.3. Radio

Ultra-wide band (UWB)-based localization uses radio waves ranging from 3.1 to 10.6 GHz. Despite the high frequency, in general, the localization has highly accurate positioning with an error level of a few centimeters. In addition, it is characterized by low power consumption due to its row duty cycle [9]. Dabove et al. analyzed the accuracy of UWB-ToF-based localizations using multiple anchors [18] placed in several areas in a room and a tag, and the accuracy of the positioning was within a few tens of centimeters. Wang et al. proposed a robust localization method using UWB for NLOS (non-line-of-sight) environments. The localization method integrates UWB, an inertial navigation system (INS), and a floor map [19]. To mitigate the noise in the UWB observations, they introduced an adaptive robust extended Kalman filter algorithm. The proposed method achieved accuracy corresponding to a root mean-square error of 0.27 m in a building that was several tens of meters-squared in size. Queralta et al. proposed a UWB-ToF-based localization method for UAV positioning [20] using multiple anchors placed at several areas in a room and estimated the position of a moving UAV equipped with a tag with an error of a few tens of centimeters. Krishnan et al. analyzed the accuracy of UWB-TDoA-based localizations using multiple time-synchronized anchors placed in a room and a moving robot, achieving an error of less than 25 cm [21]. The main drawback of this localization technology was that the typical localization range using UWB-ToF and UWB-ToA was a couple of ten meters [22]. Therefore, estimating the long-range positions of CNs is difficult.

Wi-Fi-based localization techniques are also popular in indoor localization. A CN of our sewer inspection system has a Wi-Fi module for sending video data to an AP. If Wi-Fi-based localization is available for the sewer pipe environment, there is no need for additional sensors on the CN, reducing the cost of implementation and weight limitations. Wi-Fi-based localization using radio signal strength (RSSI) is popular. Poulose et al. conducted localization experiments based on Wi-Fi fingerprinting by using a smartphone device and multiple Wi-Fi RSSI measurements and achieved localization errors within 0.44 m for a linear motion device in an indoor environment [23]. Hashem et al. proposed an indoor localization system that combined fingerprinting and Wi-Fi-ToF-based ranges with respect to IEEE 802.11mc. In the localization system, multiple Wi-Fi RSSI measurements installed on one floor of the office estimated the position of a mobile device. They achieved an accuracy of 0.77 m [24]. Kim et al. conducted an RSSI-based ranging experiment with IEEE 802.11n at 2.4 GHz in an indoor environment and achieved localizations of up to 10 m [25]. However, radio communication ranges in narrow pipes are restricted by frequencies. As described in Section 1, Nagashima et al. confirmed that the wireless communication range with IEEE 802.11n at 2.4/5 GHz was 10 m inside a 200 mm in diameter sewer pipe. They also found that the available range in the lower radio frequencies—such as 920 MHz used by ARIB STD-T108—is much shorter in narrow sewer pipes.

Maletic et al. conducted a performance evaluation for mmWave-ToF-based ranges in the 60 GHz band in an indoor environment. In the experiment, two anchors placed 1.8 m apart estimated the position of a tag in a room within an error range of 5 cm [26]. In addition, Nisshinbo Micro Devices Inc. developed an mmWave ranging sensor available in Japan called NJR4652 [27]. They achieved a detection angle of objects in a horizontal direction of no more than 90 degrees [28]. However, since the CN of our sewer inspection system rotates while drifting in water, using the directional antenna to localize the CN is difficult.

### 2.4. SLAM

Aitken et al. [29] comprehensively surveyed SLAM (simultaneous localization and mapping) techniques, including landmark-based localization, that can be used by self-propelled robots moving in sewer pipe networks. Some of the techniques introduced in the paper would be able to be used in our scenario. However, most of these localization techniques require the installation of an encoder to measure tire rotation speed and an IMU in addition to the camera, which increases the weight and the computational cost of the machine for the localization. Our paper, on the other hand, proposes a method of localization with a drifting device with a monocular camera and a Wi-Fi module, and we demonstrated it in an underground pipe.

### 2.5. Inertial Measurement Units

Inertial measurement units (IMUs) can potentially be used with our sewer inspection system to localize CNs. IMUs aided by odometry sensing have been used to localize mobile robots [30,31]. Murtra et al. proposed an odometry-based localization of a mobile robot based on the fusion of sensor data of an IMU and cable encoder, which measured the length of an unfolded cable from the starting point of operations up to the tethered robot [30]. Al-Masri et al. proposed the odometry-based localization of a mobile robot using the fused sensor data of an IMU and encoder [31]. Yan et al. proposed a real-time localization and mapping system utilizing measurements from wheel odometry, an inertial measurement unit (IMU), and a tightly coupled visual-inertial odometry (VIO) in GPS-denied complex greenhouses [32]. These methods combine inertial sensors with encoders that measure rope lengths and tire rotation speeds in order to navigate the robot. These methods were implemented in environments where the robot was traveling stably on the ground. However, in our system, wireless CNs drift over water and rotate. Thus, cable encoders cannot be used, and the rotation of the CN leads to significant errors. In addition, mounting additional sensors is difficult because of the weight constraint.

### 2.6. Cameras

The CN used in our sewer inspection system was equipped with a camera for inspection. If a CN can be localized with a camera module alone, there is no need for the additional cost of mounting another sensor on the CN. Hansen et al. proposed a localization method that uses a visual localization simultaneously with mapping (vSLAM) with a monocular camera and stereo camera [33,34,35]. However, the patterns of the captured frames in sewer pipes are often uniform in nature. Therefore, accurately tracking changes in the singular points reflected in the frame is difficult. Hence, the longer the device travels, the larger the localization error. To reduce the error, Zhang et al. [36], Alejo et al. [37]. and the authors of [38] proposed methods for localizing the detecting landmarks, such as sewer pipe joints and utility holes, by image processing to correct long-term localization errors. These methods either mount cameras in front of self-propelled robots or use multiple cameras. However, in our scenario, tracking the singular points using the front camera is difficult because the CNs may rotate during drifting. Moreover, damages are often found at the upper side of the pipe. Thus, the camera should be mounted on top and not in front of the CN, and it records a video of the top of the pipe. In addition, many sensors mounted on the CN increase the weight of the CN and make it difficult to drift. Thus, additional devices should not be mounted on a CN. Then, we designed a localization method that depends on only a fish-eye lens camera mounted at the top of the CN.

## 3. Sewer-Pipe-Inspection System with Drifting Wireless Cameras

In this section, we provide an overview of the sewer inspection system with drifting wireless CNs that we developed for sewer pipes with diameters of 200 to 250 mm, which are the most popular pipes in Japan. We inspect several hundred meters to several kilometers at a time. A CN and an AP communicate with each other via wireless LAN, and their communication range is about 5–10 m, which is much shorter than the distance between utility holes, which is typically 30 m or more.

### 3.1. Basic Design

Figure 1 shows an overview of the system. The inspection flow of the system is as follows. First, an inspector places APs in multiple utility holes in the inspection section. Then, the inspector inserts multiple CNs into the sewer pipe from a utility hole at appropriate intervals. After being inserted into a pipe, they start recording the inside of the pipe. When each of the CNs enters the communication range of the wireless LAN of an AP, it transmits the video data to the AP. Then, the AP, in turn, forwards the video data to the video server via a cellular network. Finally, the inspector accesses the video server and views the videos recorded by multiple CNs. Since the inspector need not enter the pipe and wait until the CNs reach the end of the inspection range to view the video, this system reduces sewer inspection labor costs. Generally, by using a self-propelled camera robot in a sewer pipe, the pipe needs to be cleaned after stopping sewer water flows. However, our system does not require stopping the water flow, and it does not require cleaning the pipe. Therefore, the inspection operation can be performed safely and quickly. Moreover, the inspector can respond quickly to problems such as the failure of CNs and stalling in the pipe. In addition, when severe damage is discovered with respect to a pipe, the inspector can quickly address the failure.

Each CN consists of a camera, sensor, light, wireless LAN interface, battery, and a small computer installed in a waterproof transparent capsule. We developed a prototype CN for this system. Figure 2 shows a photograph of the prototype CN.

Each AP shown in Figure 3 was equipped with a transceiver, a battery, antennas, and a computer unit. The APs were installed inside utility holes and communicate with CNs drifting inside the pipe via wireless LAN. They also communicate with the server in the cloud and outside the utility holes via a cellular network and transfer videos received from the CNs.

### 3.2. Transmission of Video Data from a Drifting Wireless Camera to an AP

As described in Section 1, the communication range of IEEE802.11n wireless LAN in a small diameter sewer pipe is quite short. Therefore, the amount of video data transferred from a CN to an AP can be quite small. Let us assume that a CN drifts at a speed of 0.3 m/s and that the CN stores the video at a bit rate of 2.5 Mbps, frame size of 1920 × 1080, 30 fps, and in H.264 format; and that APs are installed at intervals of 200 m in a sewer pipe. Let us also assume that the video data transfer rate over a wireless LAN is 20 Mbps and that the communication range of the AP is 8 m, which is the distance from which a CN can communicate with an AP while moving within a range of 16 m. Under these conditions, the CN accumulates approximately 200 MB (2.5Mbps×200m/0.3m/s/8bit) of video data while drifting between two neighboring APs, say $AP_{k-1}$ and $AP_{k}$ in Figure 4. The CN takes approximately 100 s (=200MB/20Mbps) to transmit the 200 MB video data to the downstream AP ($AP_{k}$). Since the CN drifts at 0.3 m/s, the period of time in which the CN is within the communication range of the AP is about 53 s (=16m/0.3m/s). This duration is insufficient for transmitting all video data. The CN cannot transmit all video data captured between APs within the interval of communicating with the AP. Therefore, Yasuda et al. proposed a protocol sewer video transmission protocol 2019 (SVTP2019), which utilizes multiple CNs to transmit videos covering all sections between the APs.

### 3.3. How Multiple CNs Transmit Video Data of a Different Part of the Section Relative to an AP

SVTP allows multiple CNs to transmit videos of different parts of a section to a downstream AP based on the timestamp of each CN when they capture the videos. Figure 4 shows an example of the transmission of video data from two CNs to an AP. $CN_{i}$ is the ith CN released into a sewer pipe, $AP_{k}$ is the kth AP from the released point of the CNs, and $t_{i,k}$ is the timestamp when $CN_{i}$ first receives a beacon packet from $AP_{k}$. The following is a brief description of the SVTP. $AP_{k}$ periodically broadcasts beacon packets, including τ: the elapsed time after receiving the first beacon from $AP_{k-1}$ that corresponds to the position of the upper stream edge of a section of the sewer pipe that the AP wants to receive from the next CN. If $AP_{k}$ has never received video frames captured between the section between itself and $AP_{k-1}$, then τ=0. Precisely, $AP_{k}$ receives video frames after each CN receives a beacon from $AP_{k-1}$. For example, if $CN_{1}$ sends Δt seconds of video frames captured between $t_{1,\;k-1}$ and $t_{1,\;k-1}+\Delta t$ while passing through the wireless communication range of $AP_{k}$, then $AP_{k}$ starts to broadcasts a beacon packet, including τ=Δt. The next CN, $CN_{2}$, will send the video of the rest of the sections between $AP_{k-1}$ and $AP_{k}$. $CN_{2}$ receives a beacon packet from $AP_{k}$. The beacon packets include the updated τ=Δt. Thus, it sends video frames that it captured after $t_{2,k-1}+\Delta t-\delta t$, where δt is a margin for the overlap of the sections included in the videos sent by $CN_{1}$ and $CN_{2}$.

### 3.4. Where Was the Frame Captured?

If the velocity of the water in the sewer pipe is constant and known and the position of the CN when it first received a beacon from an AP is known, then estimating the position of the CN when it captured a video frame between the AP and the next AP is easy. However, the speed of the water flow is not generally constant, and APs are installed at only some of the utility holes in the inspection area. Thus, if the distance between APs is long, the position estimation error caused by assuming a constant water speed will be significant. In addition, even if the water flow is constant, the positions where $CN_{i}$ and $CN_{j}$ firstly receive the beacon packet from some APs differs because of the interval of beacons from the AP and the interval between the release times of $CN_{i}$ and $CN_{j}$. To minimize this error, in the following section, we propose a method to locate CNs based on the known locations of landmarks such as APs, utility holes, and pipe joints.

## 4. Linking a Video Frame to the CN’s Position

In this section, we propose a method for estimating the location of a CN based on the elapsed time after the CN passes the closest landmark. We use the linear interpolation technique based on the location of landmarks, such as APs and utility holes with known positions.

### 4.1. Problem Definition and System Model

The problem we solve in this paper involves identifying the position of a drifting camera node (CN) moving in a sewer pipe, and this is performed by assuming the following system model.

The CN cannot use a GNSS. The CN and APs do not have sensors that can be used to estimate the position of the CN except for cameras and wireless communication interfaces.The CN can communicate with access points (APs) via a radio communication link when it is close to one of them (e.g., the distance between the CN and an AP is less than 10 m). The APs are installed in some of the utility holes. The distances between neighboring utility holes are longer than twice the maximum communication distance between the CN and an AP. The received signal strength of the radio signal from an AP is sufficiently low when the distance between the CN and the AP is longer than the maximum communication distance. Thus, the CN cannot use the received signal strength of the radio signal from an AP to estimate its position when the distance between the CN and the AP is long.One camera is installed in the CN. The camera points straight up and records videos of the pipe wall, ceiling, and the interior of a utility hole that is close to the CN.The CN records videos while it drifts in the sewer pipe and records the timestamp when each video frame is captured. When the CN can communicate with an AP, the CN sends the recorded video data with timestamps of the video frames to the AP. The video server that receives all data received by the APs estimate the CN’s position when each frame is captured.The starting position of the CN and positions of all utility holes and joints of pipe segments are known.

### 4.2. Localization Strategy

As described in Section 2, ultrasonic and radio-wave-based indoor localization methods are unsuitable for estimating the location of CNs drifting in sewer pipes. Considering the cost and weight, estimating the position of CNs using only wireless LAN functionality and cameras attached to the CN is difficult without adding any new devices to the CN. In addition, the CN’s position estimation error should be less than the minimum unit relative to pipe maintenance so that it does not lead to construction work in the incorrect areas. The typical segment size of a 200 mm in diameter sewer pipe is 2 m [39]. Thus, the maximum error of the estimate of the CN’s position should be less than 2 m. Since the shapes of joints of neighboring pipe segments and utility holes are easily detected by using classical image-processing techniques, as we explain later, they can be used as landmarks to estimate the position of the CN. Therefore, the proposed method uses a camera and the wireless LAN function to detect landmarks in a sewer pipe and to estimate the position of a CN using linear interpolation based on the elapsed time after passing the last landmark.

### 4.3. CN Localization Based on Linear Interpolations Using Landmarks

For the sake of simplicity, the following description assumes that only one CN is used; the same method can be used to locate two or more CNs. The timestamp used to localize a CN is assumed to be obtained from the clock on the CN. Let $P_{m}$ be the position of the mth landmark that the CN encountered, since it started drifting, and $t_{m}$ be the time when the CN passes $P_{m}$, as shown in Figure 5. Then, the position of the CN P(t) at any time *t* during interval [$P_{m-1}$,$P_{m}$] (the linear interpolated route as shown in Figure 6 is expressed as follows.
(1)$$P\left(t\right)=P_{m-1}+\left(P_{m}-P_{m-1}\right)\frac{t-t_{m-1}}{t_{m}-t_{m-1}}$$

The equation uses only the time difference of local clocks on the same device. Therefore, the local clocks of the CNs need not be synchronized with other devices. When aggregating video data, the video server can estimate the position at which the video frame was captured based on the local timestamp of the CN linked to each frame. Since the estimation error monotonically increases with the distance between the last landmark that the CN passed and the CN, the more landmarks there are, the more accurate the estimation.

The potential landmarks in sewer pipes include utility holes, sewer pipe segment joints, and APs. The typical distance between the utility holes of small-diameter sewer pipes (ϕ200 mm–250 mm) targeted in this study is several tens of meters, whereas the length of a sewer pipe segment is 990–2000 mm [39]. Therefore, if all joints can be detected, the estimated position error of the CN by linear interpolation will be less than 2000 mm.

### 4.4. Landmark Detection

We can easily detect the positions of the utility holes and joints of pipe segments using classic image-processing techniques.

**Utility holes** Since the shape of the lid of a utility hole is a circle, we can identify the position of a utility hole by detecting a circle of a suitable size from a video frame captured by a CN, as shown in Figure 7. The time when the center of the detected circle is the closest to the center of the frame is recorded as the time when the CN passes the utility hole. A circular shape can be detected using the Hough transform technique [40].**Joints of pipe segments** Since the shape of the joint of two pipe segments is a straight line perpendicular to the pipe’s direction, we can identify the position of a pipe’s joint by detecting a linear shape from a frame recorded by a CN and checking the relative angle of the line and the pipe’s direction. The pipe’s direction can be detected by finding two dark areas corresponding to the upstream side and downstream side, as shown in Figure 8. The time when the center of the detected line is the closest to the center of the frame is recorded as when the CN passes the landmark: a pipe joint. Such a linear shape can generally be detected using the Hough transform technique [41].

The detailed operation for detecting a circle or straight-line shape is as follows. Firstly, we convert the color of each captured video frame to grayscale and apply histogram flattening processes. Then, we apply image edge detections based on the Canny method [42] implemented in openCV [43]. The Hough transform is applied to this binary image to detect straight lines and circles.
**First beacon reception points from an AP** Each AP sends beacon packets periodically. When a CN receives a beacon packet from an AP with an ID that it has not received before, it stores the current timestamp.
Figure 7Example of a video frame captured at the bottom of a utility hole.
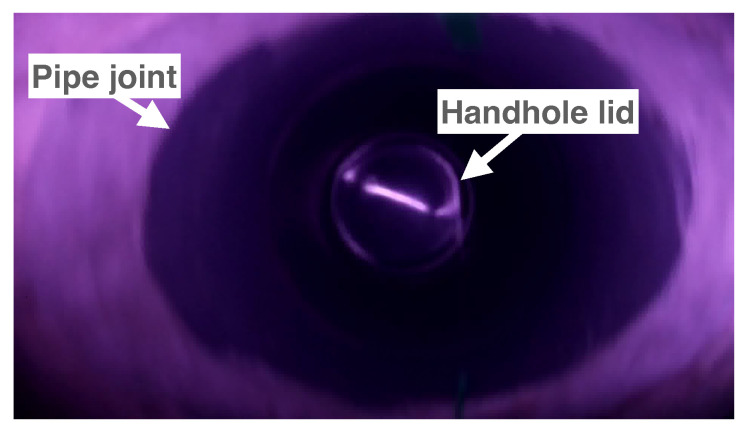

Figure 8Example of a video frame captured at a pipe segment joint.
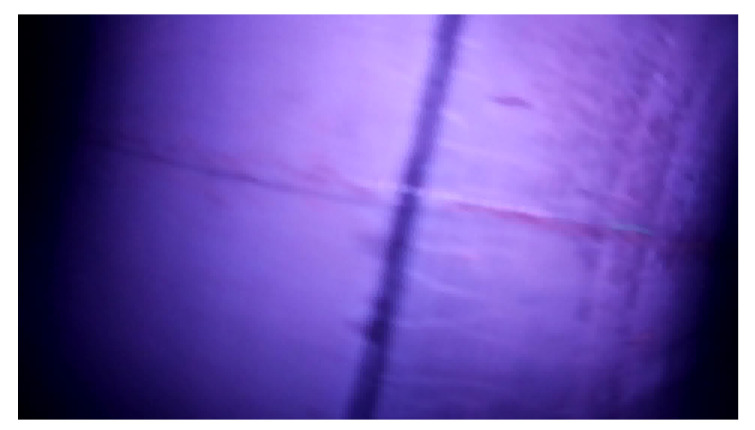


## 5. Implementation

In this section, we describe the implementation of a function for estimating the CN’s position and for linking it to each video frame, as explained in Section 4. We implemented a function for the CN’s position estimation on the video server. Video data with timestamps were sent from each CN to APs. Then, APs forwarded the video data to the video server. Finally, the video server executed the CN localization function with the received video data.

### 5.1. Implementation of a Drifting Wireless Camera

We implemented a prototype of a CN, as shown in Figure 9. The CN prototype was equipped with a Raspberry Pi 3B, a camera with a fish-eye lens, LED lights inside a plastic container, and a battery. It records videos at a resolution of 1080p, frame rate 30 fps, and in H264 while moving inside of the sewer pipe.

### 5.2. Implementation of Utility Hole Detection

The implementation of image-processing functions required for CN localization was based on OpenCV [43]. Although we proposed three landmark detection methods (utility holes, pipe joints, and APs), we implemented a function for the utility holes to demonstrate the basic performance of the proposed method. To find the shape of a utility hole’s lid in a captured video frame, the function detects a circle with a suitable radius as a utility hole lid using Hough transforms (cv2.HoughCircles( )). Figure 7 shows a frame in which a CN drifts inside a sewer pipe and captures a 200 mm diameter utility hole lid. As shown in the figure, multiple circular-shaped objects (the edge of a utility hole’s lid and the edge of the joint connecting the utility hole and the sewer pipe) were included in the same frame when the utility hole lid was detected. Since the size of the circle of the edge of the joint is clearly larger than the size of the circle of the lid, we could easily and only detect the circle of the lid based on the radius of the circle. Thus, we determined the circle’s radius for detecting the lid using the detection system based on preliminary experiments. Since multiple consecutive frames included the images of the utility hole lid, we needed to identify the best frame from them; i.e., we needed to select a frame in which the lid was located closest to the center of the frame. We selected a frame that had the timestamp closest to the average of the timestamps of frames that included the shape of the utility hole’s lid.

### 5.3. Implementation of Linear Interpolation

After the video server finds the timestamp of the moment, the CN passes the utility hole (referred to as the passing time); it links the timestamp of each frame of the received video data to the location based on the linear interpolation, as follows. Let *t* be the timestamp of the frame. The video server finds the number *m* that satisfies the following condition: $t_{m}\leq t\leq t_{m+1}$, where $t_{m}$ is the time the CN passes the *m*-th landmark. From the locations of the *m*-th and m+1-th landmarks, $P_{m}$ and $P_{m}+1$, which are provided for the video server in advance, the video server estimates the CN’s position, P(m), using Equation (Equation 1). We assume $t_{0}=0$ and $P_{0}=0$ because the first landmark is the point where the CN is inserted into the pipe. If *t* is larger than $t_{m_{\max}}$ ($m_{\max}$ is the farthest landmark number detected by the CN), then the passing time of the last landmark, $m_{\max}-1$, is given to *m* in Equation (Equation 1).

## 6. Operation Verification

We verified the accuracy of the proposed localization system using an underground pipe testbed. First, we explain the environment for the verification of the localization accuracy and verification method. Then, we show the results of the verification. Finally, we discuss the localization accuracy of the proposed method based on the results.

### 6.1. Verification

#### 6.1.1. Verification Environment

We evaluated the performance of the proposed system in an underground reinforced concrete sewer pipe on our campus, as shown in Figure 10a,b. The length and diameter were 11 m and 250 mm, respectively. The depth from the ground surface to the bottom of the pipe was 520 mm. The pipe had vertical holes every 1 m, and we set lids and placed sandbags on all vertical holes during the experiments. The pipe had two water pools at both ends of the pipe. We placed a pump at the bottom of the pool (right side in Figure 10a) to pump water to the pool at the upper side through a pipe in order to generate a water flow. The pump was adjusted to generate a flow velocity of approximately 0.26 m/s. In addition, we set a rope with a scale from the upper side of the sewer pipe (the CN insertion point) to the removal point in advance so that we could visually confirm the captured location of each video frame.

#### 6.1.2. Verification Method

We conducted the experiments using the following procedure. First, to control the CN, we pre-connected it to a PC via SSH in the 2.4 GHz IEEE 802.11n ad hoc mode. As soon as we released the CN from the first upstream handhole point, we sent a raspivid command [44] from the PC to the CN to start the camera recording. The recording duration was set to 40 s, which was longer than the time taken for the CN to drift down the sewer pipe. After the CN drifted down the pipe, we retrieved the CN. Since conducting more trials renders the measurement results reliable, we repeated this process 15 times. The results show that the 15 trials seem to be sufficient for understanding the accuracy of the proposed system.

After the trials, we transferred the videos recorded by the CN to a Linux machine, which ran the video frame localization program explained in Section 5 for each video frame and recorded the position of capture of each frame. We then checked the estimated positions of the captured video frames (via a web page we developed, as shown in Figure 11). To confirm the ground truth of the captured positions, we read the scale value in the image of the rope measure in each frame that estimated the captured positions at 0.5×nm(n=0,1,2,…,17). We then checked the positions where the frames were captured by looking at the image of the rope scale included in the frames.

### 6.2. Results

Figure 12a shows the relationship between the estimated location of the CN and the ground truth over the 15 trial experiments in our proposed method. To evaluate the effect of detecting landmarks, we demonstrate the location estimation without using landmarks except for the start point and the last handhole. Figure 12b shows the relationship between the estimated location of the CN and the ground truth when under linear interpolation with only two landmarks at the 0 and 8 m points. For each trial, we plotted the relationship from 0 to 8.5 m, every 0.5 m, as we described in Section 6.1.2. The result of each trial was depicted with the different colors on the graph. The red dotted lines represent the positions of the handholes at 1, 5, 7, and 8 m in the sewer pipe. In Figure 12a, in all the trials, there were no significant shifts in the drifting positions. Furthermore, the localization error was within the distances relative to the adjacent landmarks. Comparing Figure 12a,b, we can observe that the variation in trajectories is larger in Figure 12b.

Figure 13a shows the distribution of the position estimation errors at various positions, and Figure 13b shows the error under the condition with linear interpolation by only two landmarks (landmark with 0 and 8 m points). For each trial in the figures, we plotted the error between the estimated location and the ground truth from 0 to 8.5 m, every 0.5 m. The result of each trial was depicted with the different colors on the graph in the same manner in Figure 12. In Figure 13, the errors at the handhole locations (1, 5, 7, and 8 m) were within 0.1 m. As shown in Figure 7, the entire circular shape of the handhole (shot from the bottom of the pipe to the top) was included in the video frame. Considering that the diameter of all the handholes was 200 mm, even with a 0.1 m error, the system successfully detected the shapes of all the handholes included in the camera view. The maximum error was approximately 0.46 m at the 2.5 m point, which was almost at the middle of the longest interval of handholes.

Figure 13a presents that the longer the distance between the CN and the closest handhole, the more significant the variance in the position estimation’s error. This indicates that the velocity of the CN was affected by the water flow and the collisions of the CN with the pipe wall. The errors in the section between 1 and 5 m were all negative, indicating that the server estimated the point in front of the ground truth, as the CN may be increasing its velocity from the released point until it reaches a velocity equal to the velocity of the water flow. Therefore, we can estimate that the drifting velocities around 1 and 5 m points were slower than the average section velocity. Figure 13a has a smaller overall error in position estimations than Figure 13b. Furthermore, the maximum error in Figure 13a was −0.46 m at the 2.5 m point, and Figure 13b had a larger error of −1.15 m at 3.5 m and 4.0 and 4.5 m points. It can be seen that the more landmarks our proposed method uses for linear interpolation, the smaller the error in position estimations.

Figure 14a shows the cumulative distribution function (CDF) of the estimated position error observed at all estimated positions for the 15 trials. Figure 14b shows the CDF under the linear interpolation with only two landmarks at 0 and 8 m. In Figure 14, more than 90% of the estimated drifting position errors were within 0.3 m. On the other hand, Figure 14b shows that more than 90% of the estimated drifting position errors were within 0.9 m.

### 6.3. Discussion

It was observed that the estimated drifting position errors were within the distance between neighboring handholes, which prevented the possibility of inspecting and excavating the incorrect section of the sewer pipe.

In 2-meter-long sections, between 5 and 7 m and between 7 and 8 m, the localization error was within 0.15 m, as shown in Figure 13. Since the length between the joints of sewer pipes is typically 2 m, as described in Section 4, it can be said that if both utility holes and the joints of the sewer pipe are added as landmarks, the error will be less than 0.15 m.

In the experiment, the CN detected all handholes in all trials. However, if some of the landmarks are not detected in the inspection area, it is impossible to use consecutive landmarks to estimate the CN’s position between them, causing a significant error during linear interpolations. To avoid errors, a function for detecting landmark detection failure is needed. As the typical joint interval and the time of passing the utility holes are known, and the moving speed of the CN can be estimated from the history of the time of passage, the landmark detection failure can be easily detected. However, the CN may collide with obstacles in a pipe, e.g., tree roots, and temporarily stop, causing a significant error in linear interpolation. To tackle the drawbacks of linear interpolation, we propose adding IMU and the odometry function described in Section 2 to the CN to reduce the estimation error at locations far from landmarks.

In these experiments, the drifting speed of the CN was set to 0.26 m/s. However, in a real sewer pipe environment, a CN drifts at a variable speed. If the speed is fast, the video frames captured by the CN cannot reflect all scenes of the inspection area. Let us consider a case when a CN captures video frames at 30 fps, one video frame covers a sewer pipe of length ranging 0.2 m (according to Section 6.2), and the shutter speed is sufficiently high for neglecting the drifting speed of the CN; then, the drifting speed of the CN is vm/s. The CN should satisfy condition v≤0.2m/f×30fps=6m/s to capture the entire scene of the inspection area. If the CN captures at 60fps, it should satisfy condition v≤0.2m/s×60fps=12m/s as well. Considering the possible water speed in sewer pipes, this requirement for the velocity of the CN is easily achievable.

## 7. Conclusions

This paper described the proposal, design, and implementation of a video frame localization system for a drifting-camera-based sewer-pipe-inspection system. The main contributions of this paper are summarized as follows. Firstly, we proposed a method for estimating CN positions based on linear interpolation using utility holes and pipe joints as landmarks in a sewer pipe. We also implemented the proposed method together with a browsing system that links location information to each frame of the video and conducted experiments to evaluate the accuracy of the proposed localization function using a drifting CN in a real underground pipe. The evaluation results show that the utility holes (handholes) included in the video frames captured by a camera on a drifting camera node (CN) were successfully detected by using the Hough transform technique and contributed to accurately estimating the position of the CN, i.e., the position where each video frame was captured. The proposed localization system only requires the minimum required components on a CN, a camera, a light, and a Wi-Fi module, which are originally needed for the drifting-camera-based sewer-pipe-inspection system.

The results showed that all handholes, installed at 1, 2, and 4 m intervals, were successfully detected, and the maximum localization error was less than 0.46 m, which was 11.5% of the maximum interval of the handholes. Assuming that the pipe joints are used as landmarks, the accuracy of the video frame’s localization with the proposed method will be enough to identify damaged pipe segments, since the intervals are typically 2 m long. To the best of the authors’ knowledge, this is the first study presenting the localization of a drifting camera in a narrow sewer pipe and not a self-propelled robot, and it is based on image processing.

In the future, we plan to (1) achieve accurate detection of pipe joints; (2) conduct experiments using actual sewer pipes with large utility holes (manholes), obstacles, and damage; (3) combine the proposed method with IMU for more accurate localization of CNs; and (4) propose a method to correct the localization of CNs in cases some of the landmarks are failed to be detected.

## Figures and Tables

**Figure 1 sensors-23-00793-f001:**
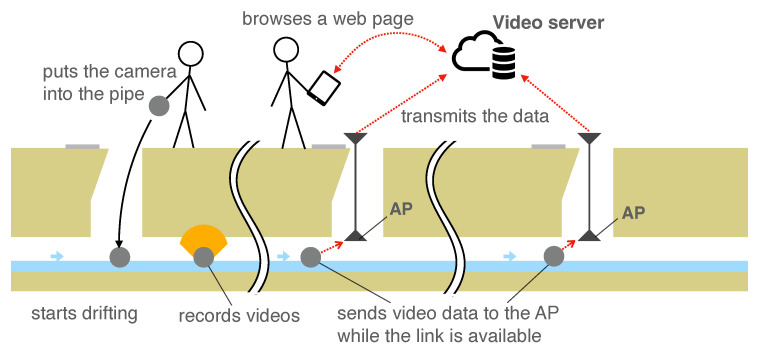
Sewer inspection system using drifting wireless camera nodes: An inspector places the drifting wireless camera nodes (CNs) into the pipe. Then, each CN starts drifting and recording videos inside the sewer pipe. Each CN sends video data to access points (APs) when it is within the wireless communication range of one of the APs. The AP receives the data and forwards the received data to the video server. Users can access the server and identify damaged points of the pipe via web browsing.

**Figure 2 sensors-23-00793-f002:**
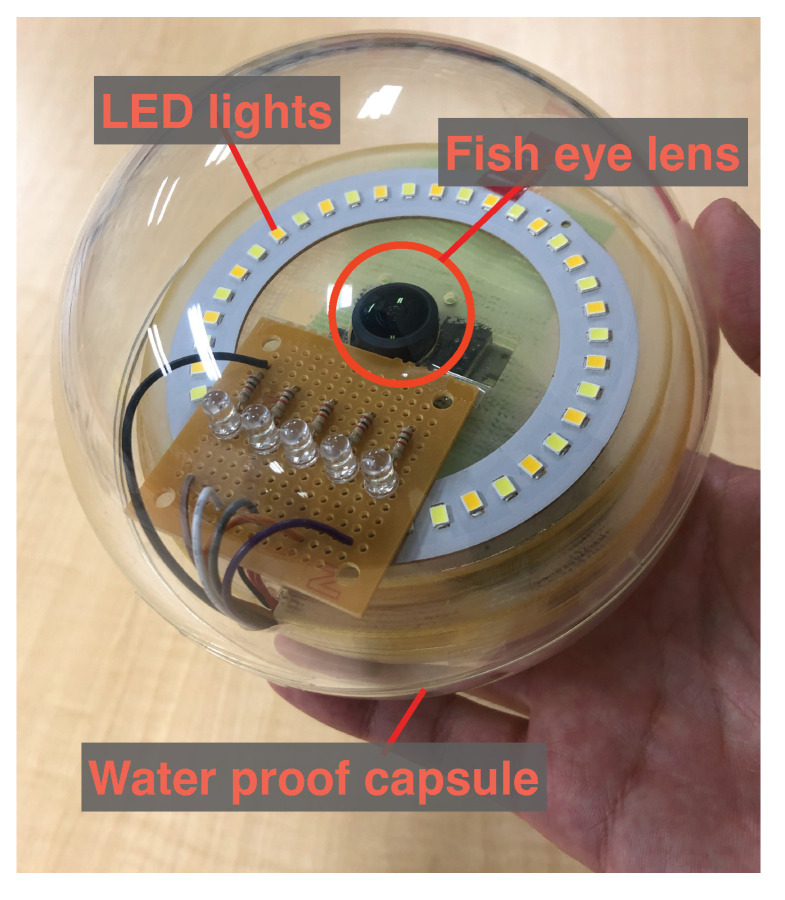
CN prototype with a waterproof capsule.

**Figure 3 sensors-23-00793-f003:**
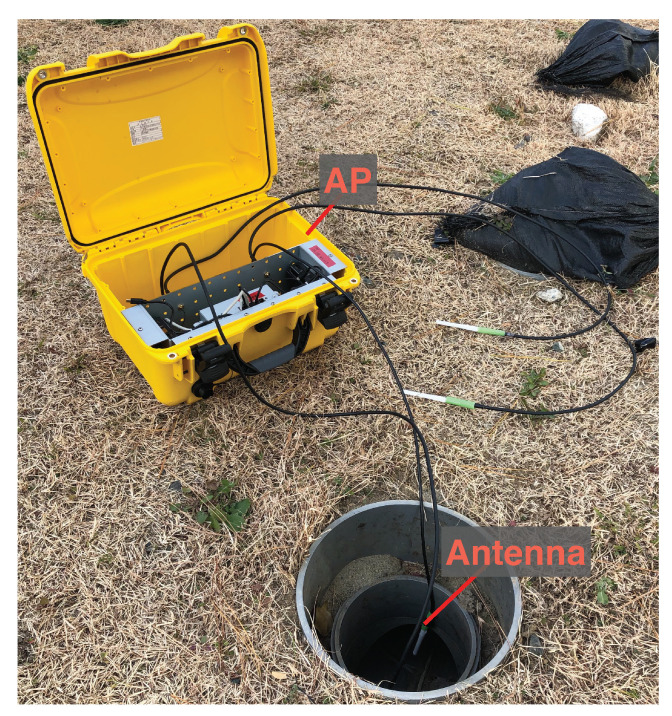
An access point installed in a hole.

**Figure 4 sensors-23-00793-f004:**
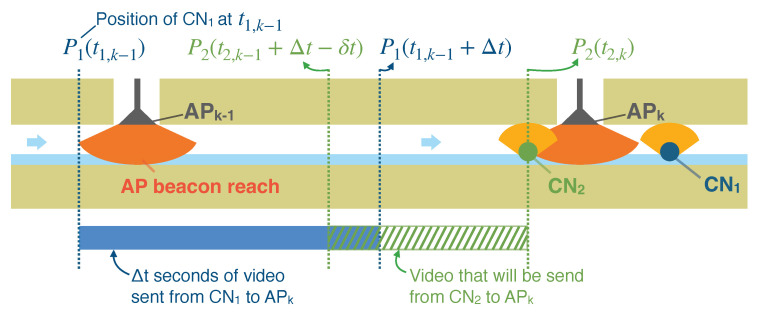
Example of transmission of video data from multiple CNs to $AP_{k}$.

**Figure 5 sensors-23-00793-f005:**
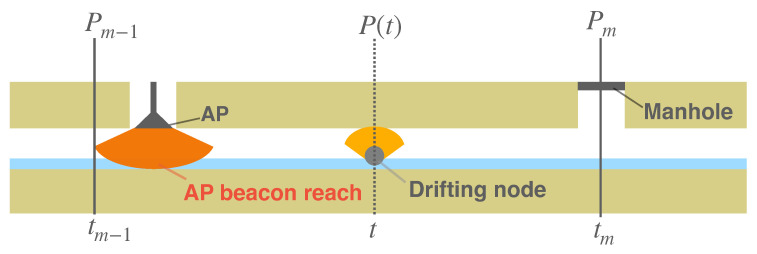
Landmarks (AP and utility holes) and a CN drifting between them.

**Figure 6 sensors-23-00793-f006:**
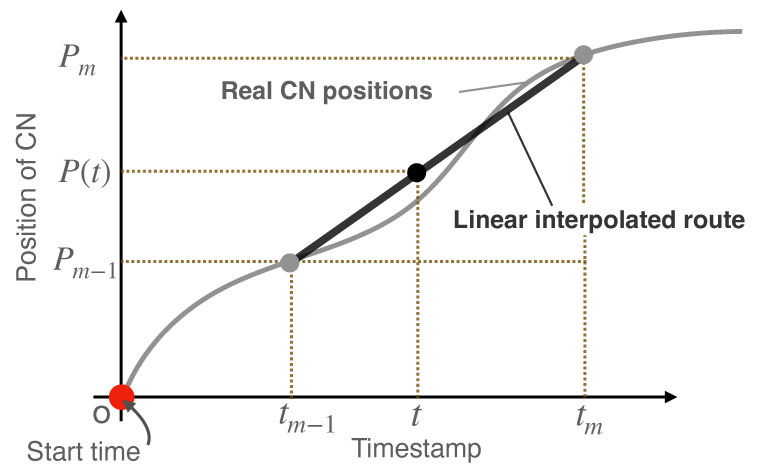
Linear interpolation using landmarks.

**Figure 9 sensors-23-00793-f009:**
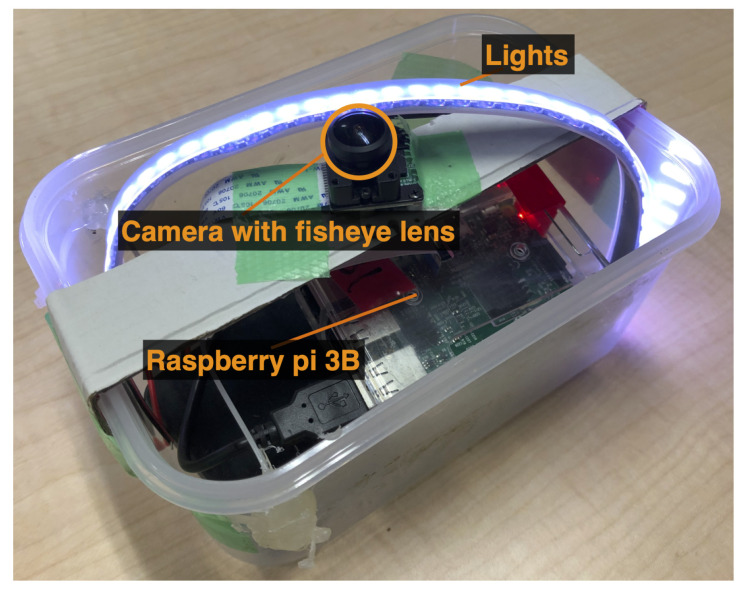
A prototype CN.

**Figure 10 sensors-23-00793-f010:**
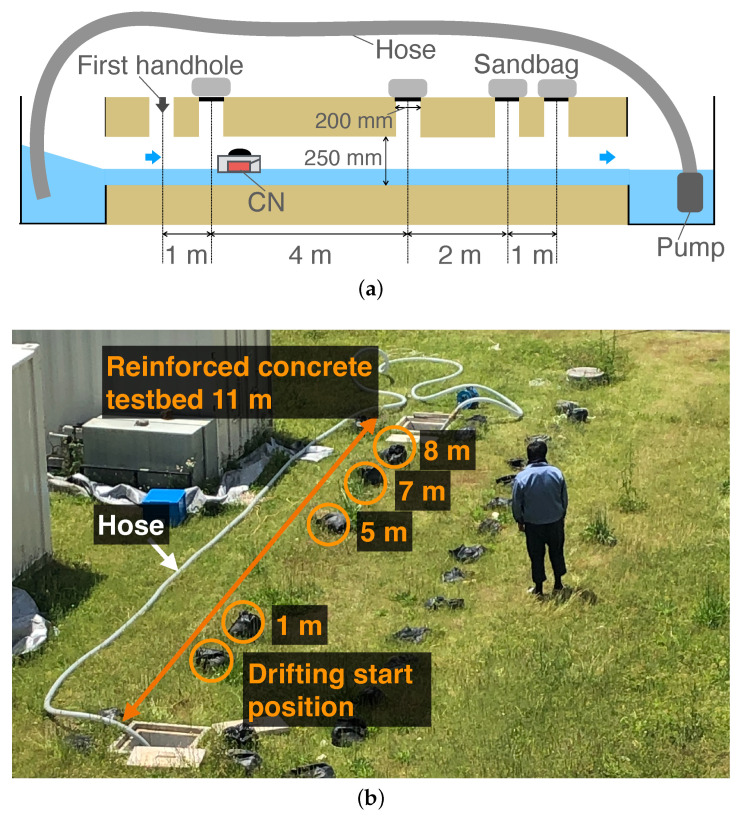
Overview of the testbed. (**a**) Structure; (**b**) Picture.

**Figure 11 sensors-23-00793-f011:**
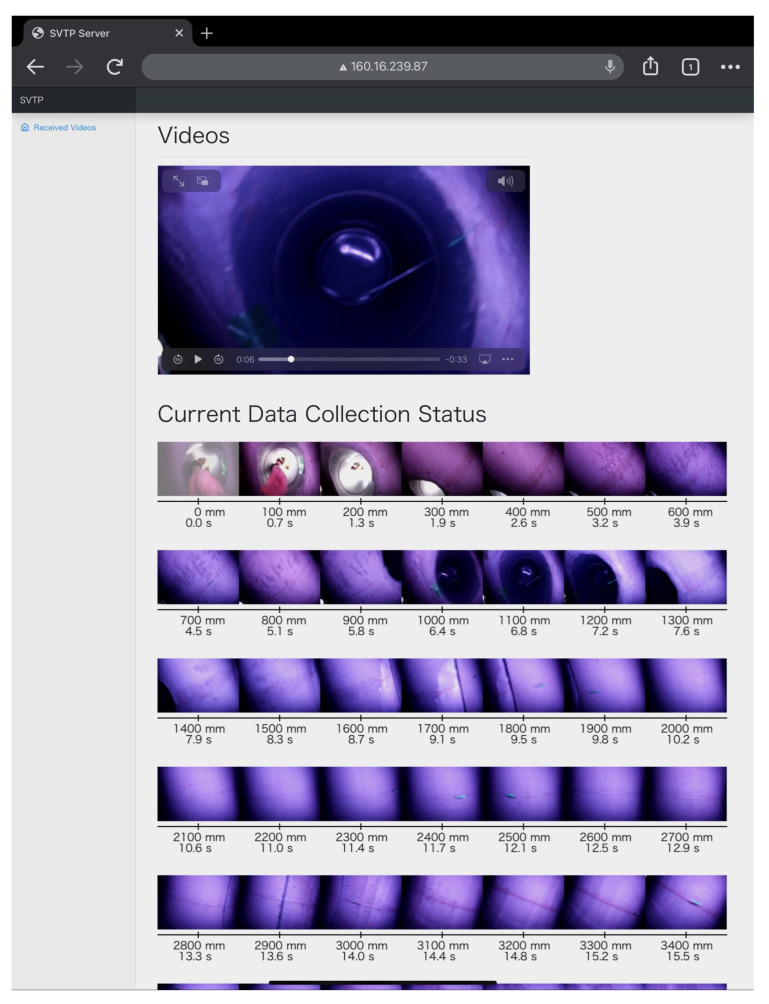
Web page for comparing the video frames and the captured position (displaying a result of the first trial experiment).

**Figure 12 sensors-23-00793-f012:**
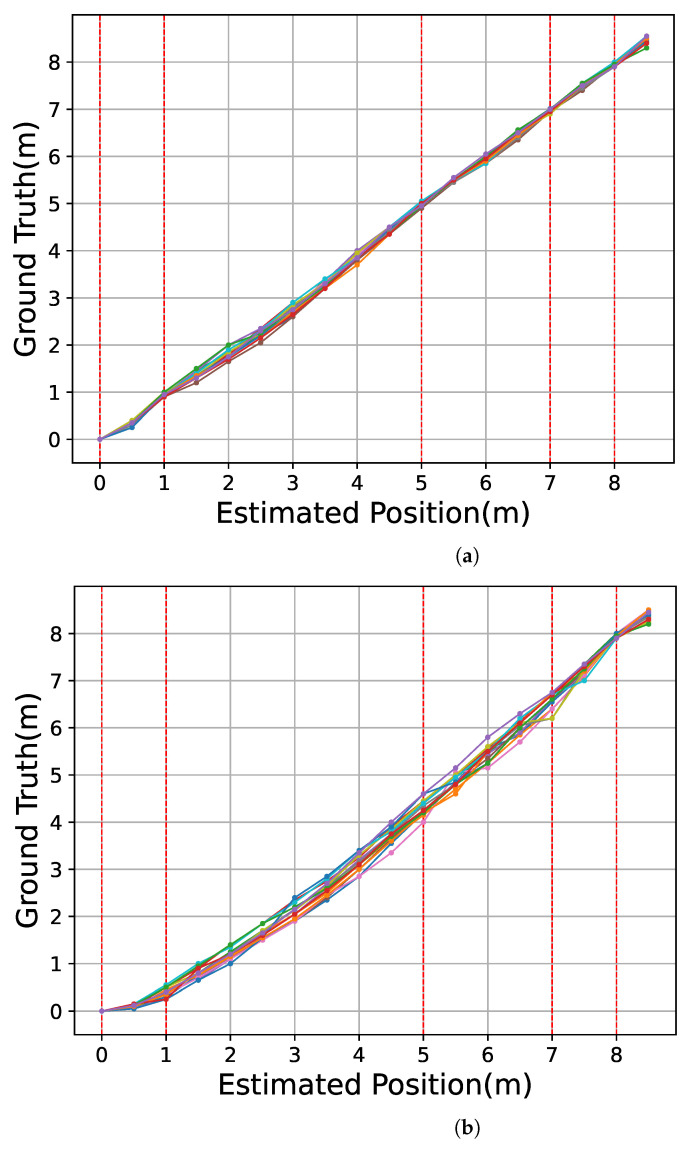
Relationship between the ground truth and the estimated captured position of the video frame under the linear interpolation. Each color corresponds to each of the 15 trials. (**a**) Landmarks at 0 m, 1 m, 5 m, 7 m, and 8 m are used; (**b**) Landmarks at 0 m and 8 m are used.

**Figure 13 sensors-23-00793-f013:**
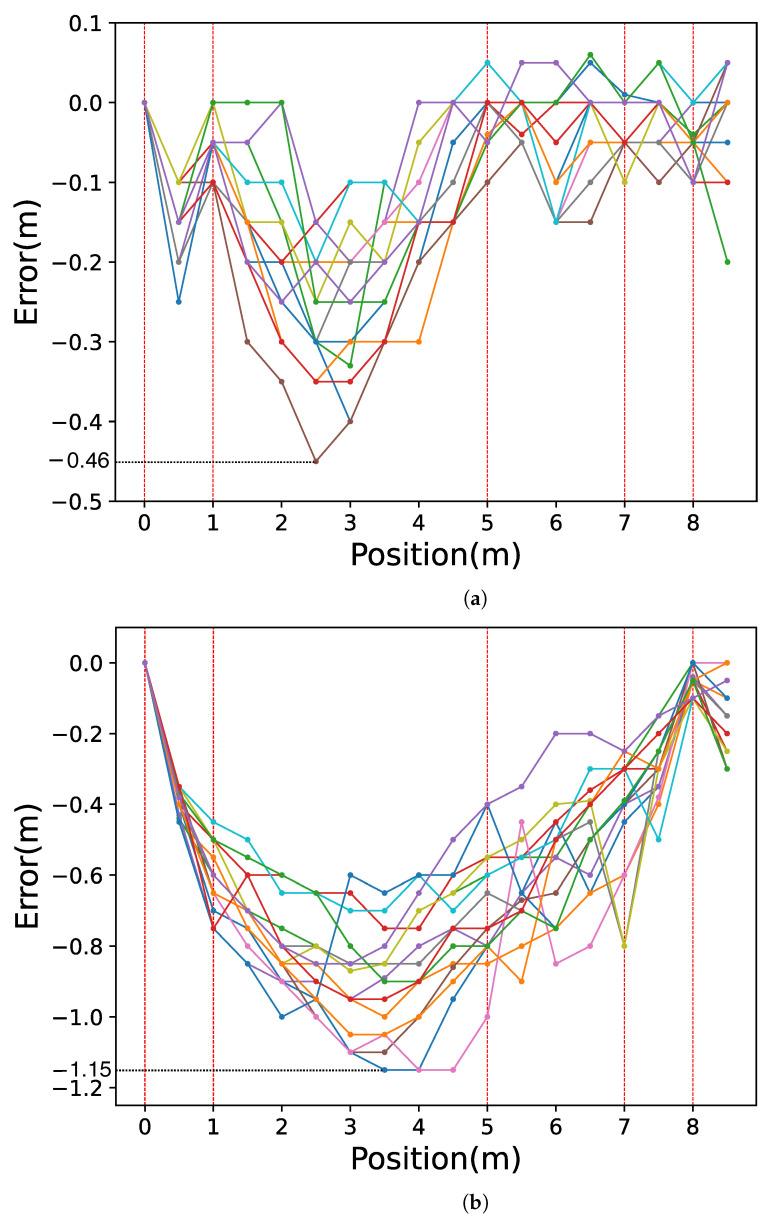
Video frame estimated position error. The error between estimated location and ground truth is depicted with different colors for each of the 15 trials. (**a**) Landmarks at 0 m, 1 m, 5 m, 7 m, and 8 m are used; (**b**) Landmarks at 0 m and 8 m are used.

**Figure 14 sensors-23-00793-f014:**
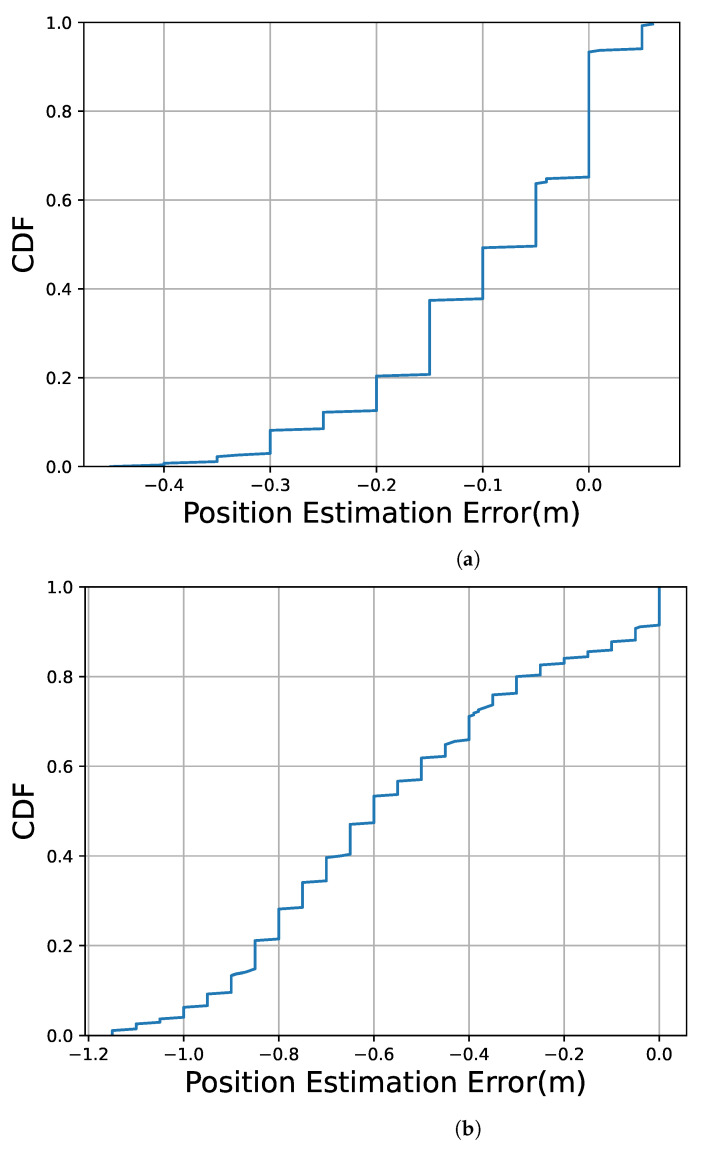
CDF of errors in the video frame’s estimated position. (**a**) Landmarks at 0 m, 1 m, 5 m, 7 m, and 8 m are used; (**b**) Landmarks at 0 m and 8 m are used.

## Data Availability

Not applicable.

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
