# Peer review of "Design and Implementation of a Video-Frame Localization System for a Drifting Camera-Based Sewer Inspection System"

_sensors, 2023, doi:10.3390/s23020793_

Round 1

Reviewer 1 Report

In order to reduce the cost of inspecting old sewer pipes, the authors worked on developing a low-cost sewer inspection system. The system uses a drifting wireless camera that records video of the interior of the sewer pipe as it drifts. An existing challenge with this task is that in small diameter sewer pipes, it is difficult to identify drift nodes throughout the pipe using Wi-Fi based positioning due to limited radio wave coverage and the unavailability of GPS. In response, the authors used image processing techniques to localize the pipes by associating each video frame with a location. Experiments show that all utility holes are successfully detected as landmarks with a maximum location estimation accuracy of 11.5% below the maximum interval of the landmarks. The design and presentation of the system in this paper is very adequate.

The following are my comments.

1. There are not enough comparisons in the experimental part of the paper, it needs to be enriched as much as possible.

2. There are many long sentences in the paper, which makes it dry to read, please try to use short expressions.

3. The logic among languages needs to be strengthened;

Reviewer 2 Report

On the other hand, the paper should be revised by considering the following issues:

MAJOR ISSUES

+ Introduction section should be improved to give the motivation more clearly.

+ The main contributions of the paper should be clearly given as a separate subsection in the introduction section.

 + The organization of the paper should be clearly given as a separate subsection in the introduction section.

+ The number of references are sufficient. Most of the references in this paper are mostly recent publications (within the last 5 years) and relevant. On the other hand, the bibliography should be improved by adding most recent references.

+ Section “Problem Definition and System Model” should be provided clearly as a separate section.

+ The proposed scheme performs well. The motivation behind it should be explained better.

+ Preamble information between section "6. Operation Verification" and subsection "6.1. Verification" should be improved.

+ The figures/schemes are generally clear. They show the data properly. It is not difficult to interpret and understand them. On the other hand, Figure 1 should be explained better by adding more information to its caption.

+ Figure 12 and Figure 13 includes many lines; however, they do not provide any legends related to those lines. The authors should explain the meaning of each line (which scheme each of those lines correspond to).

+Why are 15 trials chosen for Figure 14?

+ Section "6. Operation Verification" should be improved. Figures should be clearly explained, especially in the text/main body of the paper.

+ The conclusion should be improved by giving the key results and main contributions more clearly.

+ Future work part should be given in the conclusion section.

 MINOR ISSUES

+ The grammatical errors and typos should be fixed.

+ Size of Figure 11, 12, 13, 14 should be increased.

+ The references in the bibliography should be given in the same style. The following link should be checked: https://www.mdpi.com/authors/references 

Round 2

Reviewer 2 Report

The paper is revised considerably based on my comments on the previous version of the paper. On the other hand, the paper should still be revised by considering the following issues:

MAJOR ISSUES

+ Introduction section should be improved to give the motivation more clearly.

+ The related work of the paper should be improved by adding more referencesSome other physical layer security methods like RF fingerprinting techniques can perform very well for security attacks against the cyberphysical systems. What is the motivation of the proposed approach in this paper? For this purpose, I strongly recommend the authors should include the following papers in their related work in order to clarify not only the main contribution but also motivation of this paper in the related literature.

- C.Comert, M.Kulhandjian, O.M.Gul, A.Touazi, C.Ellement, B.Kantarci, and C.D'Amours. 2022. Analysis of Augmentation Methods for RF Fingerprinting under Impaired Channels. In Proceedings of the 2022 ACM Workshop on Wireless Security and Machine Learning (WiseML '22), pp. 3–8. https://doi.org/10.1145/3522783.3529518 

G. Reus-Muns, D. Jaisinghani, K. Sankhe and K. R. Chowdhury, "Trust in 5G Open RANs through Machine Learning: RF Fingerprinting on the POWDER PAWR Platform," GLOBECOM 2020 - 2020 IEEE Global Communications Conference, 2020, pp. 1-6, doi: 10.1109/GLOBECOM42002.2020.9348261.

+ Most of the references in this paper are mostly recent publications (within the last 5 years) and relevant. On the other hand, the bibliography should be improved by adding most recent references.

+ Section “Problem Definition and System Model” should be provided clearly as a separate section.

+ Preamble information between subsection "6.1.Verification" and subsubsection "Verification environment" should be improved.

+ The figures/schemes are not clear. They do not show the data properly. It is difficult to interpret and understand them. On the other hand, Figure 1 should be explained better by adding more information to its caption.

+ Figure 12 and Figure 13 includes many lines; however, they do not provide any legends related to those lines. The authors should explain the meaning of each line (which scheme each of those lines correspond to).

+ Future work part should be improved in the conclusion section.

 MINOR ISSUES

+ The grammatical errors and typos should be fixed.

+ The references in the bibliography should be given in the same style. The following link should be checked: https://www.mdpi.com/authors/references 

Author Response

Firstly, we would like to thank you for your valuable review comments. We believe your comments greatly helped us improve our manuscript. We would like to reply to your comments and describe how we modified the manuscript in accordance with your comments below.

MAJOR ISSUES

1. Introduction section should be improved to give the motivation more clearly.

We revised the section "1.2 Motivation"  as follows.

Even when video data can be reliably collected via wireless communication at the neighborhood of APs, identifying the positions of pipe damages, such as cracks and clogged tree roots from the captured video, is difficult because it is unclear where the video was taken. To detect the positions of pipe damages by looking at the aggregated video data, it is necessary to identify where each video frame was taken. However, GNSS cannot be used in sewer pipes, and location estimations of CNs using radio waves are also difficult because the range of the radio wave’s communication in a narrow pipe (diameter of 200-250 mm) is limited. Also, the CN's weight should be light so that it can easily drift down a sewer pipe. Thus, the additional devices for the location estimation should be minimum. To tackle this problem, it is necessary to link each video frame to the position where it was captured with the minimum additional devices and without depending on radio communication.

Note that our paper does not discuss the security problem. The security problem is out of the scope of this paper. Also, it will be quite rare that malicious users place some attacker devices or eavesdropper devices in a sewer pipe, and the wireless communication range is very limited in a narrow sewer pipe. Thus, security issues related to wireless communication cannot be the main issues of our work.

2.  The related work of the paper should be improved by adding more references. Some other physical layer security methods like RF fingerprinting techniques can perform very well for security attacks against the cyberphysical systems. What is the motivation of the proposed approach in this paper? For this purpose, I strongly recommend the authors should include the following papers in their related work in order to clarify not only the main contribution but also motivation of this paper in the related literature. 

    1. 1. C.Comert, M.Kulhandjian, O.M.Gul, A.Touazi, C.Ellement, B.Kantarci, and C.D'Amours. 2022. Analysis of Augmentation Methods for RF Fingerprinting under Impaired Channels. In Proceedings of the 2022 ACM Workshop on Wireless Security and Machine Learning (WiseML '22), pp. 3–8. https://doi.org/10.1145/3522783.3529518 
    2. 2. G. Reus-Muns, D. Jaisinghani, K. Sankhe and K. R. Chowdhury, "Trust in 5G Open RANs through Machine Learning: RF Fingerprinting on the POWDER PAWR Platform," GLOBECOM 2020 - 2020 IEEE Global Communications Conference, 2020, pp. 1-6, doi: 10.1109/GLOBECOM42002.2020.9348261.

We do not agree with the comment. Our paper is not related to physical layer security methods. We do not understand why the reviewer recommends papers on physical layer security. 

RF fingerprinting can be used for the localization of mobile devices in a location where the devices can receive RF signals from fixed transmitters. However, such a technique cannot be used in the situation assumed in this paper, where a drifting wireless camera node cannot receive RF signal from access points. We have mentioned this point in Sec 2.3 in the original version of our manuscript with some references.

3. Most of the references in this paper are mostly recent publications (within the last 5 years) and relevant. On the other hand, the bibliography should be improved by adding most recent references.

We added the following descriptions with a new reference.

2.3 SLAM

Aitken et al. [29] comprehensively surveyed SLAM (Simultaneous Localization and Mapping) techniques, including landmark-based localization, that can be used by self-propelled robots moving in sewer pipe networks. Some of the techniques introduced in the paper would be able to be used in our scenario. However, most of these localization techniques require the installation of an encoder to measure tire rotation speed and an IMU in addition to the camera, which increases the weight and the computational cost of the machine for the localization. Our paper, on the other hand, proposes a method of localization with a drifting device with a monocular camera and a Wi-Fi module and demonstrates it in an underground pipe.

[29] Aitken. J.M.; Evans M.H.; Worley R.; Edwards S.; Zhang. R.; Dodd T.;  Mihaylova L.; Anderson S.R. Simultaneous Localization and Mapping for Inspection Robots in Water and Sewer Pipe Networks: A Review. IEEE Access, 2021, 9, 140173-140198, doi:10.1109/ACCESS.2021.3115981.

4. Section “Problem Definition and System Model” should be provided clearly as a separate section.

According to the first review comment, we have made a separate subsection "4.1 Problem Definition and System Model" in the first revision. Please refer to our response to the first review and the revised version.

5. Preamble information between subsection "6.1.Verification" and subsubsection "Verification environment" should be improved.

According to the first review comment "Preamble information between section "6. Operation Verification" and subsection "6.1. Verification" should be improved," we have inserted the following sentences in the preamble information in the first revision as follows.

First, we explain the environment for the verification of the localization accuracy and verification method. Then, we show the results of the verification. Finally, we discuss the localization accuracy of the proposed method based on the results.

In the second review, the reviewer commented "Preamble information between subsection '"6.1.Verification" and subsubsection "Verification environment" should be improved.' However, now we do not have any preamble information between "6.1 Verification" and subsubsection "Verification environment." We do not have information that should be improved.

6. The figures/schemes are not clear. They do not show the data properly. It is difficult to interpret and understand them. On the other hand, Figure 1 should be explained better by adding more information to its caption.

In the previous review comment, reviewer 2 wrote "The figures/schemes are generally clear. They show the data properly. It is not difficult to interpret and understand them." On the other hand, in the second review, reviewer 2 mentioned "The figures/schemes are not clear. They do not show the data properly. "The reviewer commented on the figures/schemes completely different from the previous one. If the reviewer thinks that the second review comment is what he/she wants to say, please provide the reason why the reviewer does not think the figures/schemes are not clear. We cannot revise the manuscript without the reason.

As for Figure 1, we have added the following sentences in the caption of Figure 1 in the previous revision. We hope the reviewer refer to our response letter for the first revision.

Sewer inspection system using drifting wireless camera nodes: An inspector places the drifting wireless camera nodes (CNs) into the pipe. Then, each CN starts drifting and recording videos inside the sewer pipe. Each CN sends video data to access points (APs) when it is within the wireless communication range of one of the APs. The AP receives the data and forwards the received data to the video server. Users can access the server and identify damaged points of the pipe via web browsing.

7. Figure 12 and Figure 13 includes many lines; however, they do not provide any legends related to those lines. The authors should explain the meaning of each line (which scheme each of those lines correspond to).

Each line in Figure 12 and 13 corresponds to one of the 15 trials. We have added the following sentences to the caption of Figure 12 and Figure 13 to clarify the meaning of each line in the previous revision.

Caption figure 12: “Relationship between the ground truth and the estimated captured position of the video frame under the linear interpolation. Each color corresponds to each of the 15 trials.”

Caption figure 13: “Video frame estimated position error. The error between estimated location and ground truth is depicted with different colors for each of the 15 trials.”

8. Future work part should be improved in the conclusion section.

We revised the last paragraph of Sec. 7 "Conclusions" as follows.

In the future, we plan to 1) achieve accurate detection of pipe joints, 2) conduct experiments using actual sewer pipes with large utility holes (manholes), obstacles, and damage, 3) combine the proposed method with IMU for more accurate localization of CNs, and 4) propose a method to correct the localization of CNs in cases some of the landmarks are failed to be detected.

MINOR ISSUES

1.The grammatical errors and typos should be fixed. 

We carefully read the paper and fixed grammatical errors and typos.

2.The references in the bibliography should be given in the same style. The following link should be checked: https://www.mdpi.com/authors/references 

We fixed wrongly formatted parts in the bibliography.